# The Dynamics of Climate Change Adaptation in Sub-Saharan Africa: A Review of Climate-Smart Agriculture among Small-Scale Farmers

**Victor O. Abegunde [1], Melusi Sibanda [1,*] and Ajuruchukwu Obi [2]**

[1] Department of Agriculture, University of Zululand, KwaDlangezwa 3886, South Africa; overcomers001.av@gmail.com
[2] Department of Agricultural Economics & Extension, University of Fort Hare, Alice 5700, South Africa; aobi@ufh.ac.za
* Correspondence: SibandaM@unizulu.ac.za; Tel.: +27-(0)-35-902-6068

**Abstract:** Climate-smart agriculture (CSA) as a credible alternative to tackle food insecurity under the changing climate is gaining wide acceptance. However, many developing countries have realized that concepts that have been recommended as solutions to existing problems are not suitable in their contexts. This paper synthesizes a subset of literature on CSA in the context of small-scale agriculture in sub-Saharan Africa as it relates to the need for CSA, factors influencing CSA adoption, and the challenges involved in understanding and scaling up CSA. Findings from the literature reveal that age, farm size, the nature of farming, and access to extension services influence CSA adoption. Many investments in climate adaptation projects have found little success because of the sole focus on the technology-oriented approach whereby innovations are transferred to farmers whose understanding of the local farming circumstances are limited. Climate-smart agriculture faces the additional challenge of a questionable conceptual understanding among policymakers as well as financing bottlenecks. This paper argues that the prospects of CSA in small-scale agriculture rest on a thorough socio-economic analysis that recognizes the heterogeneity of the small farmer environment and the identification and harnessing of the capacities of farming households for its adoption and implementation.

**Keywords:** adoption; agriculture; climate-smart agriculture; mitigation; small-scale; sub-Saharan Africa

## 1. Introduction

Climate change constitutes part of the pressing and momentous threats facing the agricultural system globally [1]. Juana et al. [2] explain that the climate trend in Africa shows an increased temperature of about 0.7 °C in the 20th century, stating a rise in the temperature trend of all sub-Saharan African regions. In the low-emission setup, the African summer temperature is projected to rise at about 1.5 °C above the 1951–1980 baseline until 2050 and maintain that state until the close of the century, while a high-emission setup in sub-Saharan Africa experiences a warming until the end of the century, attaining 5 °C above the 1951–1980 baseline by 2100 [3].

There is a decline in rainfall in the semi-arid region of the sub-Sahara and increased rainfall in East and Central Africa. These trends have been projected to linger through the 21st century, coupled with an upsurge in sea level and the frequency of drought and flood [2]. Sani et al. [4] report a 46.75-mm decline in total rainfall annually in Western Ethiopia. Sani and Chalchisa [5] indicate a 2.8 times shortage in water availability in Africa between 1970 and 1995 and a 40–60% drop in the average

discharge of rivers in West Africa. Arnell [6] forecasts increased water stress for up to 370 million people in Africa by 2025.

According to Harvell et al. [3], climate change impacts on the agricultural system will be accompanied by a population explosion and a shift in consumption patterns. The projection is that the world population numbers will increase from current levels to as high as 10 billion people by 2050 [7]. The situation calls for a significant transformation in the agricultural sector to ensure adequate food supplies for increasing demand.

A high proportion of the projected increase in the global population by the end of the century is expected to come from Africa [8]. Projections reveal that many countries in the African continent will be significantly affected by varying climatic conditions and a shift in the agricultural system [9]. Africa has been identified as a region that is profoundly affected by climate change [10]. This challenge of satisfying food demand is a more pressing issue for the continent. The high vulnerability of African agriculture to the impact of changing climatic conditions is associated with heavy reliance on rain-fed systems [11].

The small-scale agricultural sector plays a crucial role in providing food and livelihoods in many African countries [12]. Fields [13] also highlights the significance of small-scale agriculture in many developing countries. Most sub-Saharan African households rely on small-scale farming for feeding and earnings [14]. In a report released by the African Centre for Biodiversity [15], the majority of the African farmers are small-scale farmers. These small-scale farmers carry out their agricultural production on fragmented portions of land, but they remain critical to food production [16,17]. This indicates that small-scale farmers constitute a vital part of the African community.

Despite the potential of small-scale agriculture, it faces many constraints that limit its effectiveness in combating the intertwined crisis of food insecurity and poverty. Small-scale farmers are highly susceptible to the changing condition of the climate, making it one of the significant threats that small-scale farmers face [10]. Changes in climatic conditions, notably increases in temperature and patterns of precipitation, negatively influence biodiversity, exacerbate the already prevailing strain on the water supply, and worsen the vulnerability of the small-scale farming systems [18].

The reliance of small farmers on rain-dependent farming predisposes the economy of many African countries to climate change effects [19]. In many African countries, the combined effect of climate variability and poor farming practices have resulted in poor soil fertility and therefore gave rise to low farm turnout [20,21]. Furthermore, there is a solid connection between climate-related risks and the severity of poverty in rural areas [22]. The combination of these factors has worsened poverty, food insecurity, and nutrition deficiencies in many African countries.

Studies have considered two principal ways by which there can be increased food production to meet up with the increasing food demand. These are the expansion of the production area and intensification on existing croplands [23,24]. However, with increasing urbanization, land scarcity has become a serious issue, making intensification with available technologies a relatively straightforward approach to close yield gaps and bring about increased farm output [25]. By implication, the sustainable intensification of small-scale farming is a vital option to meet up with food requirements [24].

Concepts such as climate-smart agriculture (CSA) emerged to bring about an adjustment in agriculture to enhance food production while dealing with the changing climatic conditions and their increasing variability [26]. Climate-smart agriculture focuses on three main goals, which are (1) a sustainable increase in agricultural productivity to enhance income levels, food security, and development; (2) climate change adaptation and resilience from the micro to the macro level; and (3) a reduction or total removal of greenhouse emissions where possible [27]. Climate-smart agriculture as a concept enhances the resilience of agricultural systems by balancing the priorities between adaptation, mitigation, and food security [26,28,29].

There are different adaptation and productivity-enhancing strategies adopted by farmers in sub-Saharan Africa, which fall within the CSA framework [7]. Many farmers, especially those in areas with lower rainfall, have substituted crops requiring a high level of water with those requiring

a lesser degree of water for cultivation [21]. Farmers in areas with frequent flooding have shifted to short-cycle crops. Some farmers have moved to crop diversification, changing plant days and mixed cropping [21]. Climate-smart agriculture includes agricultural adaptation methods that bring about a sustainable increase in productivity. The framework of CSA aims at maximizing the benefits and minimizing the undesirable trade-offs across its multiple goals [26]. Climate-smart agricultural practices could be incorporated into a single farming system and then open up numerous benefits that can bring improvement to the livelihoods and incomes of farmers, mainly small-scale farmers [24]. Climate-smart agriculture, as a concept, is aimed at helping farmers adjust to climate change and minimize its plausible unfavorable effect on their agricultural activities and livelihoods [27]. However, past experiences have revealed that some recommended innovations have been discovered not to be applicable in many developing countries after such innovations have been adopted [30].

With the development of the CSA concept and the cultural, socio-economic, and ecological dynamics of many African countries, the adoption of CSA technologies among African farmers may encounter challenges despite the benefits it promises. Most studies have focused on climate change for African countries, including its effects and projections [31,32]. There is limited attention given to issues relating to adopting concepts or agricultural practices that can tackle climate change specifically as it relates to small-scale agriculture [33]. Looking into the assessment of the concept of CSA among small-scale farmers is essential, considering its probable food security and poverty alleviation impacts.

The scope of this paper looks at CSA within the context of small-scale agriculture. Therefore, this paper undertakes a review to assess the dynamics of the concept of CSA among small-scale farmers in African countries. The paper aims to discuss the need for CSA in small-scale agriculture in Africa, the factors that influence the adoption of CSA among small farmers in Africa, and the challenges that must be tackled for better understanding of the CSA concept to enhance its scaling up within that constituency.

## 2. Materials and Methods

This paper is a synthesis of a subset of literature relevant to CSA. Web of Science and Scopus search engines were used, with a restricted search scope due to the tendency of the search engine to give wide access to materials and intricate search terms. Gray literature of poor quality as regards policy standing such as newsletters and bulletins were purposely left out, and focus was placed on peer-reviewed scientific literature. Consideration was given to relevant peer-reviewed publications, including reviews, scientific articles, book chapters, books, and miscellaneous editorial material located in the search engines at the time of the search and dated from 2010 to 2019.

The search term "climate-smart agriculture" was combined with Africa to find literature relevant to CSA in sub-Saharan Africa. Agriculture was conceptualized as related to crop and livestock production in a very broad sense. With the search terms "climate-smart agriculture" + Africa, "climate change adaptation" + Agriculture + Africa and "climate change mitigation" + Agriculture + Africa, a total of 450 documents were retrieved from the Web of Science (228 documents) and Scopus (222 documents) search engines. Then, the retrieved documents were exported to a database in ENDNOTE X8 for screening. After the removal of duplicates, the documents were pruned to a total of 364. Based on discretion concerning relevant criteria after screening titles and abstracts, the documents were narrowed to 100 papers. The criteria used for either including or excluding literature in the study are presented in Table 1.

**Table 1.** Criteria for Inclusion and Exclusion of Literature Selected for Review.    CSA: climate-smart agriculture.

| Criteria for Including Literature in the Study | Criteria for Excluding Literature in the Study |
| --- | --- |
| Text documented in English | Text documented in languages aside from English |
| Focus is on agriculture | Emphasis is on non-farm sectors (for instance, mining and manufacturing) |
| Addresses at least one of the goals of CSA (productivity, adaptation, and mitigation) | Addresses none of the goals of CSA |
| The text contains sufficient relevant details to carry out the review | The text lacks pertinent details needed for review |

Source: Authors (2019).

*Limitation of the Study*

This paper is a synthesis of a subset of literature drawn from the Web of Science and Scopus search platforms and may not be complete coverage of literature within 2010 and 2019 time frame.

## 3. Results and Discussion

### 3.1. Climate Change and the Changing Agricultural System

Climate change is a current and pressing global challenge [28]. There have been efforts toward developing awareness that the effect of human activities on the climate has reached a state of concern and has become a threat to both physical and socio-economic systems [7]. There are engagements regarding the agents and degree of climate change, including means of addressing the risks associated with climate change, but there seems to be a high degree of consensus on the fact that the climate is changing [18]. It required studies with evidence from the Earth's surface, layers of the atmosphere and oceans, and geological transformation of historical climates to arrive at the present level of global confidence that human activity is influencing the climate [18]. It is now easy to discard the notion that human activities have no impact on the environment [7].

Climate change and variability are to a great extent a result of the increase in greenhouse gases caused by human actions [34]. Therefore, in discussing environmental issues, different vital players have viewed climate change as a significant change in the modern climate, which has been escalated by human activities [35]. Human activities, mainly burning fossil fuels alongside other activities, which include agricultural activities and other land use such as deforestation, result in a buildup of greenhouse gases [18,35]. The greenhouse gases trap the heat from the sun in the atmosphere and cause a reduction in the measure of heat escaping into space. The greenhouse gases, by trapping and re-radiating the radiation that should have escaped into the atmosphere, bring about changes in the climate system [7,35]. There is increase in temperature at the sea and land surfaces, as well as changes in sea levels, patterns of atmospheric circulation, rainfall distribution, and storm intensity [35].

The impacts of climate change have been felt globally on the natural and human systems. Recent changes that occur in the natural system as a result of climate change include soil moisture reduction in China, a decrease in tree density in the western part of the Sahel, and the retreating of glaciers in Eastern Africa and Asia [36]. The impacts of climate change on the human systems are more complex to analyze due to the influence of other socio-economic factors as well as the dynamic and context-specific nature of the changes that occur in the system [37]. Factors such as culture, livelihood choices, age, gender, education, and wealth influence the level of vulnerability, thereby causing climate change to have different impacts on different individuals [37].

Climate change affects the agricultural system in diverse ways. Many of the efforts on assessing the impact of climate change on agriculture has been on how the outputs of significant food and cash crops have been affected in different areas. Different studies have pointed out that climate change has been negatively affecting maize and wheat production, with projections that the negative impact

on production will continue [38,39]. There has been an adverse effect of climate change on rice and soybeans as well but it has been slightly lower when compared with that on maize and wheat [39]. However, positive impacts of climate change on outputs were reported in areas with higher latitude and altitude [40].

There are projections of an 8% reduction in yield by 2050 in both Southern Africa and South Asia due to climate change with a greater impact on maize, wheat, sorghum, and millet than rice, cassava, and sugar cane [41]. Variations in the pattern of precipitation in terms of timing and amount, rising radiation levels, and concentration of carbon dioxide will affect the output, development, and survival of crops [40]. Climate change also has indirect effect on biotic factors such as weeds, pests, and diseases, which in turn affect crop productivity [39].

Arable land in developing countries has been projected to decline by 110 million ha by 2080, while land that can be used for double or triple cropping in sub-Saharan Africa will likely decrease due to moisture constraints and increases in variability [40]. Increases in temperature based on projections will likely cause other varietal loss, with severe consequences with regard to genetic resources in the future [40,42]. Production is not the only important part of the agricultural activities; the harvest and postharvest phase of the agricultural cycle is equally essential. Increased temperature can enhance crop drying but can also result in the enhanced reproduction of pests and multiplication of diseases on stored produce. Postharvest losses of cereals in sub-Saharan Africa were estimated to be about $4 billion per annum [43].

Climate change also affects livestock production via heat and water stress, vectors and diseases, and the quantity and quality of feeds, systems, and livelihoods [40,42,44]. Projections of changes in temperature have been shown by different studies to likely increase food prices by 3% and 84% by 2050 [45–47]. Increases in food prices will affect many agricultural producers who are net food buyers. Climate change poses a significant threat particularly in many African countries where many households depend on agriculture for their livelihood [7]. The agricultural sectors of sub-Saharan African countries show more vulnerability [7]. The vulnerability of agricultural systems in African system countries can be attributed to the degradation of the environment and varying rainfall patterns [9].

The growing advocacy for CSA stems from the consideration that it is an efficient approach to enhance agricultural productivity under a changing climate. The adoption of CSA by farmers in sub-Saharan Africa is considered a promising approach to improve productivity among small-scale farmers and thereby enhance food security. The concept of CSA aims to jointly address food insecurity, climate change issues, and ecosystem management [26]. Therefore, it integrates social, environmental, and economic scopes of sustainable development [27]. The application of CSA evaluates economic, environmental, and social situations in a location-specific manner, to point out suitable technologies and practices that are needed for successful agricultural activities [27]. Climate-smart agriculture offers a triple-win benefit to farmers through improved productivity, enhanced adaptation and resilience, and a reduction or total removal of greenhouse gas emissions [26]. The adoption of CSA facilitates a resilient, sustainable agricultural system that can improve agricultural production under a changing climate [24].

The potential of CSA to enhance food security and resilience for the agricultural system has given it a considerable level of attention in the developing countries [27]. The concept of CSA is particularly relevant in sub-Saharan Africa where the agricultural sector, which is the most vulnerable sector to climate change, is very critical to economic development [28,48]. Given the complex nature of the agricultural system in Africa, there is an urgent need for agricultural transformation and support for adaptation decision making [48]. Therefore, there is a need to deploy resources in a manner that will optimize the triple-win effect of CSA to maximize the gains from the agricultural system in changing climatic conditions.

### 3.1.1. Climate Change Adaptation

There is an expectation that climate change would have adverse effects on agricultural production, especially in sub-Saharan Africa [49–51]. With an increase in climatic variability and shocks, there are expectations of frequent and severe floods and droughts, which will raise risks attached to crop and livestock production [42]. Climate change threatens food production and access for both rural and urban settlements in sub-Saharan African through the reduction of income from farming, increase in risks, and distortion of markets [34].

Adverse impacts from climatic shocks can be decreased through adaptation efforts, which could range from slight to significant changes in approach that can bring about transformation in the farming systems. There are different ways of building adaptive capacity, but an essential component involves creating an ecosystem within an agricultural system that enhances resilience, access to breeds of livestock, and varieties of crops that have a higher tolerance for heat, drought, and flood; improves water storage systems; and builds institutional capacity for the enhancement of collective action, the dissemination of knowledge, and undertaking local adaptation planning [52].

Several of the practices that fit into the aforementioned attributes are variants of sustainable intensification, while others such as disseminating information and strengthening capacities for institutions also support the widespread adoption of sustainable intensification [53]. However, farmers will not adopt techniques or practices to adapt to climate change without a promise of increased returns on investments from such efforts.

Studies have been conducted on adaptation strategies adopted by farmers in addressing climate change and variability in sub-Saharan Africa. Farmers in regions with lower precipitation have substituted the cultivation of crops with high water requirements with the cultivation of those with a lower level of water requirement [4,6]. Crop farmers in areas with frequent flooding have adopted planting short-cycle crops and have changed their time of planting to avoid periods with heavy rainfall [4,54].

Farmers, to adapt to climate change, adopt crop diversification, change planting days as a response to changing precipitation patterns, plant tree crops, practice mixed cropping, and generate income from alternative sources [2,33]. In Southern Africa, where there is severe water stress, farmers adopt water conservation techniques such as water harvesting, wastewater re-use, and irrigation [54,55]. West African farmers respond to short intensive rainy seasons by planting short-duration crops, adopting soil conservation, and farming upland [33]. Livestock farmers dig boreholes in dry regions, diversify their income-generating activities, and shift to livestock with a high tolerance for water scarcity and high temperature [31].

### 3.1.2. Climate Change Mitigation

Agriculture is reported to significantly contribute to global warming and is responsible for about 19–29% of the emissions experienced globally [28]. With increasing food demand resulting in the need for increased food production, emissions of greenhouse gases tend to increase, owing to a possible increase in livestock husbandry, the use of agrochemicals, and land-cover change [52].

Climate change mitigation is focused on lowering or removing greenhouse gases emissions, such as for example reducing nitrous oxide and methane emissions per unit of output [56]. Sustainable intensification on existing farmlands is a primary means of achieving mitigation through the reduction of land cover change [56]. While production methods that are less intensive with lower output can be beneficial to the immediate environment, such practices may require that farmlands are put into use elsewhere to close the gap created by the lowered production, eventually resulting in a higher overall impact [57].

### 3.1.3. Triple-Win Effect of Climate-Smart Agriculture

Climate-smart agriculture is a concept designed to guide actions toward transforming agricultural systems for effective and sustainable development and food security under climate change [27]. The concept was first raised by the Food and Agricultural Organisation of the United Nations (FAO) in 2010 as an approach for achieving food security. Climate-smart agriculture was designed to accomplish three main goals: agricultural sustainability with an improvement in productivity, resilience (adaptation effect), and a reduction or total removal of greenhouse gas (mitigation effect). Therefore, climate-smart agriculture concurrently offers the benefits of an increase in agricultural productivity, adaptation, and resilience to changes in climatic conditions while cutting back or eliminating greenhouse gas emissions [26,29].

Sustainable Increase in Agricultural Productivity

About three-quarters of the indigent global population are rural dwellers that depend mainly on agriculture for their income [58]. The World Bank [59] shows that an effective way to address concerns about food availability and the means to procure food in sub-Saharan Africa is to ensure growth in the agricultural sector. The reason is that a very high proportion of the populace relies on farming for their livelihoods. An increase in productivity coupled with cost reduction through the efficient use of resources is an essential means of achieving agricultural growth [60]. The enhancement of the richness of the agro-ecosystem offers tremendous high benefits to the producers by decreasing poverty and increasing food availability and accessibility.

Studies have revealed that except for significant technological interventions and mitigation measures, it will be challenging to achieve the projected 70% increase needed in agricultural production with the occurring environmental changes such as drought and air pollution [61,62]. However, there are current and biological means that have been proposed for enhancing agricultural sustainability and increasing food production through improved plant productivity [40]. The hormesis concept considers living materials to be dynamic and adaptable, with the potential of using low-dose stress for survival [40]. There are different means of applying the hormesis concept in the agricultural system, which could fit into the sustainable increase in productivity and food security profile of CSA and could be applicable in the agricultural system in sub-Saharan Africa.

The concept of hormesis includes the application of the enhancement of resistance to pests and micro-organisms and increasing productivity at low costs [63]. It also contains selective use for desired outcomes or attributes. Hormesis can be applied in postharvest engineering to enhance the quality and shelf life of agricultural products [64,65]. Given its potential to enhance productivity, the concept of hormesis can be considered for a sustainable increase in plant productivity and improved nutritional value, which can improve food supplies and quality. The application of hormesis could contribute an outstanding result in African agriculture, particularly in enhancing productivity and sustainability in the small-scale farming system in sub-Saharan Africa. However, there have been concerns raised on the application of hormesis in the agricultural system, which may be a challenge for the agricultural system in sub-Saharan Africa. There may be risks with the application of hormesis in an uncontrolled environment whereby irregularity in hormesis-based intervention can result in the direct opposite of the desired outcome [66]. Furthermore, it is still unclear if a higher nutritional value and quality will be obtained from agricultural products harvested from fields subjected to hormesis-based intervention [66].

Resilience to Climate Change

Work done by the Intergovernmental Panel on Climate Change (IPCC) [18] suggests that climate change impacts on agricultural production are already apparent in several areas globally. More adverse effects than positive ones have been felt by highly vulnerable African countries. Temperatures are projected to continue to rise, ushering in higher average rainfall, with wet regions and seasons

getting wetter and dry ones getting drier [24]. Reports from the IPCC [18] further reveal that the frequent occurrence of events such as drought, heavy rainfall, and flooding poses a significant threat to food availability.

However, alleviating the adverse effects of climate change is not an impossible feat but requires actions toward evolving effective and efficient adaptation strategies [24]. Given the peculiarity of the effects of climate change in a different region, coupled with the variation in agro-ecologies, effective and efficient adaptation strategies will vary for different locations [18]. Some potential adaptation measures have been identified; these include the adoption of agroecology principles and landscape approaches that enhance the resilience of agro-ecosystems and diversify production through cultivating crops and raising livestock that are stress-tolerant [22,27].

Sub-Saharan Africa is highly diverse in its agro-ecology. Therefore, there is no single universal agricultural technology or practice that can achieve climate resilience and agricultural sustainability in the region [22]. The concept of CSA in its approach identifies suitable agricultural technologies and practices for any agro-ecology [27]. This implies that there are diverse pathways to CSA, which include conservation agriculture, livelihood diversification, improved agro-climatic information, improved agronomic practices, and storage. One of the main goals of the pathways is to make the agricultural system more resilient to climate risks and shocks [67]. Climate-smart agricultural technologies and practices are more effective, featuring an efficient use of energy and genetic resources, means of water and soil management, and enhanced agricultural resilience and productivity in a changing climate [26,27]. Many of the CSA technologies and practices modify agricultural activities in ways where there is reduction in the effect of weather on agricultural production and greenhouse gases emissions. These technologies and practices help build a resilient agri-food system in sub-Saharan Africa, which is characterized by the ability to function under increasingly climatic, economic, and social shocks [67].

Reduction of Greenhouse Gas Emissions

According to the IPCC [18], agriculture is a crucial driver of greenhouse gas emissions. Its contribution to emissions principally is through crop and livestock management, as well as its activities that result in deforestation. Non-carbon dioxide emissions from agriculture are expected to accumulate as a result of anticipated growth in the agricultural sector. However, there are several means by which agriculture-contributed emissions can be reduced. A critical approach is reducing emission intensity through sustainable intensification. The approach involves employing new practices that boost the efficiency of use of input such that the increase in output outweighs that of the emissions [68]. Another approach to reducing emissions is increasing the carbon sequestration capacity of agriculture. Agro-forestry and the reduction of soil disturbance are two means of carbon sequestration in agriculture. Plants and soils have the ability to remove carbon dioxide from the atmosphere and store it in their biomass; this process is called carbon sequestration [18,69].

As a concept, climate-smart agriculture does not attempt to provide mitigation or adaptation benefits in isolation. Climate-smart practices have been reported to be enriched with the potential to simultaneously increase resilience to climate change, bring down greenhouse emissions, and improve food security [27,29]. Agricultural practices that could concurrently deliver beneficial adaptation, mitigation, and food security outcomes would be preferable to other practices for agricultural production [56].

There are different techniques obtainable for land management, including ploughing the soil, fertilizing crops, and feeding livestock. Some of these farming techniques are better than farmers' present approach to agricultural production as regards greenhouse gas emissions, resilience to changing climatic conditions, and level of output [27]. Examples of CSA practices are agro-forestry, the fertilization of farmland with livestock manure, conservation agriculture, and planting resilient crops [70].

### 3.2. The Need for CSA in Sub-Saharan African Agriculture

Over the years, attempts to improve food productivity have focused on a technology-push approach, with the assumption that a substantial increase in productivity could be quickly brought about by technologies [71]. However, technologies and concepts are conveyed to farmers without a full grasp of the local context the farmers operate from, thereby not paying adequate attention to important factors that influence successful adoption and implementation [24].

Many governments and organizations have invested hugely in agriculture through climate adaptation projects such as irrigation schemes, which include the provision of improved agricultural packages to boost the productivity of small-scale farmers [72]. However, the majority of these projects have little achievement because they are only technology-driven, with the assumption that technology development alone will suffice to drive agricultural intensification [24].

Approaches are shifting gradually from being just technology-oriented to being more system-oriented, putting into consideration the complexity of farming systems [21]. Studies have revealed that when there is a sole focus on technologies and concepts while working on agricultural innovation, the enabling and constraining factors that determine the availability, accessibility, and productive outcomes of such technologies or concepts are overlooked [71,73,74].

It has been argued that addressing climate change adaptation issues, particularly at a regional scale, calls for a collaborative approach and active networks among researchers, policymakers, and practitioners [75,76]. The collaborative approach should also involve the public and private sectors engaging the decision-makers [76]. The collaborative approach should be driven by an interdisciplinary research team constituting of members from different backgrounds and sectors of society aiming at solving complex problems at the interface between the society and the environment [77]. However, the success of an interdisciplinary approach for addressing climate change adaptation issues often relates to the level of occurrence of coevolution of understanding among the players and the degree to which the players integrate their efforts in knowledge production and application [76,77].

An example of a collaborative approach to address climate change issue is the collaborative adaptation research initiative in Africa and Asia (CARIAA), which integrates the physical, ecological, and socio-economic dimensions in its approach [76]. In its design, CARIAA adopted a hotspot approach, prioritizing the glacier-fed river basins and the semi-arid areas of Africa and Asia [78]. These areas and the inhabitants are highly vulnerable to climate change and its effects [79]. Climate change hotspots are areas where a strong indication of climate change impact combines with a large proportion of vulnerable, poor, or marginalized populace [76,78]. The hotspot approach emanates from acknowledging that climate change will not have the same level of effect on people [76].

The concept of CSA gives a more holistic approach to addressing climate change issues in the small-scale farming system of developing countries, particularly in sub-Sahara Africa [80]. Climate-smart agricultural practices include proven effective agricultural practices or techniques that agree with the three main pillars of the concept: improved productivity, adaptation and resilience, and the lowering or removal of greenhouse gas emissions [81]. Such agricultural techniques include agroforestry, mulching, intercropping, crop rotations, integrated crop–livestock management, conservation agriculture, and improved water management. Climate-smart agriculture also involves adopting and using innovative practices such as weather forecast and climate risk insurance, among others [82].

The concept attempts to bolster the links between local, national, and global stakeholders in agriculture through the promotion of adaptation and mitigation synergies across scales with simultaneous consideration of their trade-offs [26,83]. There are two essential reasons for the synergies between adaptation and mitigation as far as the concept of CSA is concerned. The first reason is that adaptation seems more appealing to farmers than mitigation, and as a result of this, adaptation as one of the CSA pillars serves as the point of entry for farmers to adopt practices that can offer both adaptation and mitigation benefits. The second reason is that a reduction in greenhouse gas emissions

is likely to occur if a broad and far-reaching application of adaptation practices with simultaneous mitigation benefits takes place [84].

The susceptibility of African countries to climate change and variability has been highlighted by different studies [85,86]. Farmers, especially small-scale farmers, which are the majority in most African countries, have to adjust and cope with variation and unpredictability in the patterns of precipitation. Despite the unpredictable pattern, small-scale agriculture is still primarily rain-fed. In addition to the varying climatic condition, an increase in the frequency of run-off and soil erosion has deteriorated many arable lands [87]. The challenges the farmers are facing with climate change have necessitated the adoption of concepts or innovations that can offer them the opportunity to improve soil productivity and mitigate climate-related risks [22].

Studies have revealed that many farmers are aware of the adverse effects of climate variability [86,88–90]. Farmers' observations have been established to be consistent with scientific assertions from empirical evidence [33]. There has been a projection of 5% drop in rainfall but a 1.5–2.3 °C rise in temperature by 2050 [18,32]. There have also been reports of reduction in crop yields and livestock production as negative consequences of climate change [1,91]. Empirical evidence suggests that there is a need to deal with climate-related risks in agriculture if many African countries are to achieve sustainable development goals [31]. The adoption of CSA is seen as a viable intervention since the concept increases productivity to improve food security, helps farmers adapt better and become more resilient to climate change, and lowers or removes greenhouse gas emissions [26].

Small-scale farmers can also play noteworthy roles in mitigating against climate change. Several specific practices across diverse agricultural systems reduce the carbon footprint and therefore play strong mitigation roles [92–94]. However, it is hard to sell mitigation as a goal to small-scale farmers [24]. Adaptation benefits that promise a speedy and direct return on investment can serve as an entry point, given that both adaptation and mitigation are complementary pillars of CSA, leading to synergistic co-benefits [95,96]. The benefits of climate-smart agriculture that have been analyzed from different literature sources are summarized in Table 2.

**Table 2.** Benefits of CSA in Sub-Saharan African Agriculture. IPCC: Intergovernmental Panel on Climate Change.

| Benefits of CSA | Literature Sources |
|---|---|
| Sustainable increase in agricultural productivity | Hansen et al. [22]; Totin et al. [24]; Lipper et al. [26]; FAO [58]; Mills et al. [61]; Springmann et al. [62]; Mathews et al. [80]; Makate et al. [81]; Murray et al. [82] |
| Resilience to climate change | IPCC [18]; Hansen et al. [22]; Totin et al. [24]; Lipper at al. [26]; FAO [27]; Were et al. [67]; Makate et al. [81]; Murray et al. [82]; Duguma et al. [96] |
| Reduction of greenhouse gas emissions | IPCC [18]; Hansen et al. [22]; FAO [27]; Thornton et al. [29]; Wollenberg et al. [56]; Smith et al. [68]; Field [69]; Pye-Smith [70]; Makate et al. [81]; Jarvis et al. [84]; Smith et al. [92]; Descheemaeker et al. [93]; Rakotovao et al. [94] |

*3.3. Contribution of Climate-Smart Agriculture to Sub-Saharan African Agriculture*

Desertification and land degradation are severe threats in the sub-Saharan region of Africa, with hunger and poverty as constant prospects. Farmers, in response, are devising means to manage forests, rangelands, and other natural resources in a sustainable manner. A range of projects are assisting farmers' efforts in tackling desertification problems and helping them improve their farmland. The goal of these projects is to advance food security by helping the farmers enhance food and timber production and build higher resilience to climate change. Reforestation helps in climate change adaptation by lowering the wind speed and thereby minimizing the harms that could be done to crops. The practice also provides a chance for carbon dioxide sequestration and climate change mitigation.

For example, in Niger, local farmers, through actions that protect and manage the natural regeneration of trees and bushes on their land, have greened about 5 million ha of land, thereby resulting in the most significant transformation of the environment in the Sahel, and perhaps in the whole of Africa [97]. Farmers have planted about 200 million additional trees on cultivated fields, yielding about 60 or more trees where it used to be 2 or 3 per hectare years back. The benefits in the region of US$56/ha per annum or a gross value of US$280 million per annum have been suggested concerning soil nutrient, fodder, food, and fuelwood [97]. There is an extra yield of 500.000 tons of cereals from these fields, catering for about 2.5 million people. This has brought about a considerable improvement as farmers only need to plant once, compared to the three or four times they used to plant years back [97].

The West Africa Agricultural Productivity Program (WAAPP) is a project funded by the World Bank, and it is aimed at transforming agricultural production to become more climate-smart across 13 West African countries, which include Ghana, Liberia, Senegal, Mali, Guinea, Sierra Leone, and The Gambia [98]. The aim of the project is to ensure sustainability in agriculture. About 160 climate-smart crop varieties were distributed, and farmers were trained in climate-smart practices, including agroforestry and composting [98]. According to a report from the World Bank [98], more than seven million farmers have been assisted through WAAPP. The project also has lower greenhouse gas emissions and increases the productivity and resilience of over four million hectares of land. Productivity has expanded by about 150%, and food production has risen by more than three million tons.

Nyando in Kenya is ravaged with soil erosion, resulting in deep gullies in the valley of the area. About 81% of families in the villages of Nyando report 1–2 hunger months or periods in a year when they cannot produce food from their farmlands. To respond to these challenges, many farmers have started adopting CSA, including the use of climate-smart species and cultivars [98]. Many farmers now adopt water conservation techniques, better livestock management using more resilient livestock varieties, better manure management, and agroforestry. Farmers have been able to improve their farms and make them more productive and high yielding. They have a better understanding of diversification, which has made their farms more resilient to climate change and lowered their carbon footprint, alongside an increase in productivity and improvement of the quality of their soils [98]. The climate-smart practices have enhanced the food security and resilience of farmers and options for adapting their agriculture.

Agriculture is a significant driver of Tanzania's economy. It accounts for about 46% of the country's gross domestic product [98]. Better performance in the agricultural sector can boost the incomes of the poor. Growth in the agricultural sector is primarily influenced by factors such as access to inputs, access to credits, infrastructure, and irrigation, among others. Irrigation provides reliable access to water, which shields farmers from the periodic shocks that emanate from climatic variability [98]. The Agricultural Sector Development Project (ASDP) includes support for irrigation investments to increase productivity and protect farmers from shocks caused by climate change. The main objective of ASDP is to enhance agricultural productivity, farm incomes, and food security. About 228,000 farmers have profited from the project, one-quarter of whom are female. Land area under irrigation has increased by over 80%, resulting in increased rice productivity, rising from about 4.5 metric tons to 5.8 metric tons [98].

The most extensive export product in Uganda is coffee, generating about 20% of the country's foreign exchange earnings [99]. However, the challenge of climate change is a threat to production and therefore, the country's economy. To respond, farmers have realized that they must adapt to the changing conditions [99]. Through the planting of shade trees, the microclimate can be changed, and the temperature in the coffee-growing areas reduced by 2–5 °C. These shade trees can generate about 50% additional revenue for farmers, absorb carbon in the soils, and reduce drought problems. Crop losses that have been averted because of the use of shade trees have been estimated to exceed more than US$100 million per year [98].

Farmers in Malawi are having difficulties with crucial trade-offs between resource use sustainability and meeting their short-term needs [97]. Therefore, support to farmers must be beneficial to them in manifold, featuring low-input and low-labor techniques alongside yield increase and soil protection. Organizations such as Total LandCare and the International Maize and Wheat Improvement Center (CIMMYT) are among those with significant prior work in the area of conservation agriculture practices to address these needs. The aim of the conservation agriculture practices is the increased productivity and profitability of small-scale farms while also enhancing how resilient they are to climate change [97]. The results from trials have shown higher and more stable yields than what is obtained from conventional ridge tillage systems. For example, returns from maize production are 11–70% higher with conservation agriculture, especially during periods of low rainfall [97].

Makate et al. [81] assess the impact of the adoption of drought-tolerant maize as CSA on the livelihood of small-scale farmers in rural Zimbabwe. The findings from their study reveal that there was a significant improvement in maize productivity and the livelihood outcomes of farmers as a result of the adoption of drought-tolerant maize. The adoption of drought-tolerant maize by the farmers brought about an increase in the yield of maize, increase in maize output per capita, and therefore an increased quantity of maize apportioned for household consumption and market sales. Table 3 summarizes CSA applications and projects in some areas of sub-Saharan Africa.

**Table 3.** CSA Application and Projects in Certain Regions of sub-Saharan Africa.

| Region | Literature Sources |
|---|---|
| West Africa | Nyasimi et al. [97]; World Bank [98] |
| East Africa | World Bank [98]; Jassogne et al. [99] |
| Southern Africa | Makate et al. [81]; Murray et al. [82]; Nyasimi et al. [97]; Mango et al. [100]; Schaafsma et al. [101] |

The adoption of small-scale irrigation farming as a climate-smart agriculture practice in the Chiyanya triangle of Southern Africa brought about an increase in the income of farming households in the area [100]. Mango et al. [100] attributed the increased revenue to increases in production as a result of the increased capacity for intensification and diversification through the adoption of small-scale irrigation. The farmers could produce off-season, as they were able to augment their crop production during dryness or water scarcity.

The contribution of CSA to Africa's small-scale farming system should not be underestimated. Small-scale farmers have enjoyed multiple benefits from the adoption of CSA and the implementation of CSA projects. However, the implementation or adoption of CSA does not come without challenges. In cases where there is limited access to finances and other support services needed for implementing and adopting new or improved agricultural practices that can enhance productivity, the likelihood of the successful implementation of CSA projects or CSA adoption may be low [82]. For instance, female farmers in the small-scale farming system of Malawi have limited access to productivity-enhancing inputs and resources. They lack not only capital, but other resources such as animal power and credit as well [82]. This affects their capacity to embrace CSA. The question arising is how can CSA be reachable to these farmers for climate change adaptation, and how can CSA be locally adapted to be applicable in their situation(s)? Other challenges identified with CSA implementation and adoption are the lack of technical expertise among small-scale farmers, as well as extension workers and development practitioners [101].

The adoption of CSA by small-scale farmers requires some level of capacity, consistency, and trade-offs from the farmers, which should be given significant consideration [82]. Whether farmers are ready to accept the required capacity and trade-offs should also be considered. For instance, conservation agriculture requires ploughing methods that reduce the extent of soil disturbance or turn over. Farmers need high management skills and will have to deal with high initial costs of specialized planting equipment to practice conservation agriculture [70]. The concept of CSA advocates for the

substitution of the use of chemical fertilizers with the use of manure, which is not without trade-offs. The use of organic manure requires a more significant use of land, less feed for livestock, and high manure management skills [70]. However, Mwongera et al. [48] opine that small-scale farmers would have more interest in the immediate benefits of agricultural production than giving much attention to any long-term technical benefits.

*3.4. Adoption of Climate-Smart Agriculture by Small-Scale Farmers in Sub-Saharan Africa*

There are different ways by which farmers in sub-Saharan African countries, especially the small farmers with limited resources, attempt to deal with climate change and variability [102,103]. Even at that, there are concerted efforts toward the development, deployment, and scaling up of climate-smart agricultural practices to enhance farmers' adaptation to changes in climatic conditions [26]. An example is the target set by the Africa Climate-Smart Agriculture (ACSA) to see to the adoption of CSA by 25 million farmers come 2025 [82].

Climate-smart agriculture encompasses agricultural practices that could avail the triple-win benefits that are the pillars or main objectives of the concept: a sustainable increase in productivity, an enhancement of resilience of livelihoods and ecosystems, and the reduction or removal of greenhouse gases [55]. Examples of such agricultural practices include conservation agriculture, agro-forestry, integrated crop–livestock management, mulching, intercropping, crop rotation, an improvement on water management, and development on grazing. Climate-smart agriculture also includes an improvement of weather forecasting, early warning systems, and insurance for climate risks [82]. In essence, CSA attempts to deliver existing practices and technologies that have been proven in practical terms into farmers' hands coupled with driving innovations such as drought-tolerant crop varieties to cope with the pressure of climate change [54].

The adoption of CSA among small-scale farmers cannot be viewed as homogenous. The implication of this is that, in order to drive the adoption of CSA among small-scale farmers successfully, there is the need to acknowledge its salient heterogeneous attribute while developing or understanding the small-scale farming system in many areas, especially in the adoption of innovations and techniques and engaging productive inputs [54]. This is because the structures of small-scale farmers differ at both the micro and macro levels, while their drivers and barriers are also different. This implies that small-scale farmers are not at the same level of resource access, land constraints, and market orientation.

For about 30 years now, studies have been focused on developing and promoting low-cost technologies that are suitable for small-scale agriculture in sub-Saharan Africa [101]. However, in recent times, the focus has shifted to addressing challenges of decreasing productivity, poor soil fertility, degrading farmlands, food shortage, and increased level of risks from agricultural production aggravated by changes in climatic conditions. Now, there is more attention to promoting proven innovations and technologies that could help adapt to the changing climate [82]. There are several technologies today that fit into this profile. Examples include crop diversification, cereal–legume intercropping, stress and drought-tolerant crops, agro-forestry, and conservation agriculture [100].

However, despite the promising potentials of these practices to improve productivity, promote livelihoods, increase ecosystems' resilience, and cut back or eliminate greenhouse gas emissions, the adoption rate among small-scale farmers is still low [104–106]. Failure to acknowledge and factor in the heterogeneity among small-scale farmers could be a limiting factor in promoting the adoption of CSA [107]. The assumption of a homogenous system in the small-scale agricultural sector while developing and scaling up CSA practices could pose a severe constraint to sustained adoption.

Data generated by the World Bank, FAO, and International Fund for Agricultural Development (IFAD) [108] indicate that knowledge base, resources, and capacity are very crucial requirements in adopting new CSA practices. Therefore, the heterogeneous qualities of the small-scale agricultural system as regards resource control and access and an array of socio-economic attributes should be considered while designing, delivering, and diffusing the technologies and practices in question. Acknowledging and taking account of the heterogeneous qualities of the small-scale farming system

give a better insight into the benefits or challenges of CSA adoption [53]. Furthermore, to facilitate the rate of CSA adoption and how effective it can be on the livelihoods of households, there is the need to continually promote the practices coupled with a comprehensive analysis of the socio-economic dynamics, which factor in the different attributes of the small-scale agricultural sector [82,109,110].

There are suggestions on the artificial stratification of small-scale farming households into smaller clusters based on homogeneity or specific criteria. Examples include classifying based on the same resources base, livelihood, opportunities, or limitations [107]. The stratification gives rise to what is called farm typologies [54]. The outcomes obtained from analyzing farm typology can help foster a more coordinated and tailored approach to agricultural development [111].

Studies on farm typology can be very resourceful in enhancing the effective and efficient implementation of CSA techniques in small-scale agriculture. Chikowo et al. [112] point out that farm typologies are crucial in comprehending and unfolding the diversity existing among small-scale farmers. They further state that analyzing farm typology can be very helpful in the impact-evaluation of climate-smart techniques on farm productivity, the resilience of ecosystems, and livelihoods. There is a need to understand the reasons behind the early or substantial adoption of CSA techniques by some farmers, and farm typology can help with that. It can be implied from the understanding of whether the adopters have significant resources (in terms of capital), a particular ability or capacity for adoption (in terms of education or social networks), or they want to try new ways of doing things [82].

Makate et al. [54] studied the adoption of CSA practices in small-scale farming systems of Southern Africa. They found that socio-economic factors such as gender, education, marital status, farm size, farming experience, frequency of extension visit, availability of labor, and involvement in other income-generating activities aside farming are active players in the adoption and use of CSA practices. Their analysis showed that these factors could form the basis for stratifying farmers as regards farm typology and that female household heads, youths with low farming experience, poor male household heads, and farmers combining other activities with farming activities appeared to be low adopters of CSA practices.

In contrast, households with extensive farmlands and labor, those into full-time farming, and wealthy male farmers who are married turned out to be better adopters of CSA practices. Their findings also show that farmers who do not recognize farming as their principal source of livelihood might not be serious with the adoption of CSA as the need to increase their farm productivity and resilience may not be pressing to them. They further state that inexperience, economic hardships, and a lack of assets hinder youths in making worthwhile farming investments; thereby, the adoption rate of CSA among them is low [54].

Onyeneke et al. [113] show that factors such as the farmers' level of education positively influence CSA adoption. The finding agrees with those of Onyeneke and Nwajiuba [114] and Onyeneke et al. [115]. Education is hypothesized to open access to information and the technical know-how needed for improved technologies [116]. A further look at the factors influencing farmers' decisions to adopt CSA shows that farmers' income contributes positively toward their decision regarding CSA adoption. This could be because high-income earning farmers have more access to resources and information and are less risk-averse [117].

Furthermore, access to credit increases a farmers' likelihood of adopting CSA practices [117]. Gbegeh [118] came up with similar findings that point to a positive association between credit access and the level of adoption. Gbegeh [118] opined that the adoption of an innovation or technology could hinge on the availability of credit since technology or innovation adoption more often than not requires capital, whether owned or borrowed. Access to extension services is another factor that positively influences the likelihood of CSA adoption, as it enhances the adjustment of production systems and the management of knowledge. Onyeneke et al. [113] reveal that exposure to extension services increases the level of awareness of farmers about climate change and agricultural techniques that could be adopted to adapt and mitigate against the challenge. This concurs with the conclusions of

Arya et al. [119] that training and enhanced information access considerably promote the acceptance of CSA.

*3.5. Challenges for CSA Adoption and Up-Scaling in Sub-Saharan Agriculture*

Despite the contributions and benefits of CSA in sub-Saharan African agriculture, CSA adoption and up-scaling are faced with some challenges. To start with, the definition of CSA by the FAO [120] is widely accepted by agricultural stakeholders. However, there are still dents of uncertainties concerning which practices should be taken as CSA practices, and also which of the three benefits offered by CSA should be prioritized in a particular context [33]. As there is growing advocacy for infusing CSA into policy frameworks addressing agriculture and rural development, there is the need to fully comprehend the context-specificity of CSA to be able to launch initiatives that can enhance the adoption and facilitate the up-scaling of CSA.

Furthermore, with the ever-increasing growing population in Africa and the pressure of food demand on the agricultural system, the agricultural system may broadly embrace intensification, which can result in activities that potentially escalate the emission of greenhouse gases. As farmers will be engrossed with improving their products to meet the food demand, policymakers and governments would have to work hard for a win–win situation [33]. There will be a need for multi-stakeholder approaches in influencing new knowledge and decisions that can significantly impact the food system and the agricultural sector. Hitherto, agroforestry and conservation agriculture are the most celebrated practices that have been deemed to fall in line with the objectives of CSA. However, the economic implications of these practices are understudied, especially with small-scale farmers. Besides, these two practices have been reported to face many challenges with adoption [121].

A high level of insight with innovations is required for the sustainability of the agricultural productivity of small-scale farmers in sub-Saharan Africa, considering the predictions of more frequent droughts and varying patterns of rainfall [18,33]. Many areas are subjected to soil fertility limitations besides stress from water shortages. Approaches that give room for multi-dimensional solutions to the problems of water and soil fertility deficiencies with opportunities for infrastructure development may broaden the possibilities for CSA solutions [33]. Considering that CSA as an approach avails the benefits of climate change adaptation and mitigation jointly, sub-Saharan African countries will have to mainstream CSA in their agricultural or environmental policies, programs, and strategies. Williams et al. [122] state that many countries south of the Sahara have not integrated climate change adaptation into their plans for agricultural investments. This necessitates the need for an improvement on CSA policies.

According to Taylor [123], there is advocacy for a paradigm shift in the agricultural system with CSA, without adequate consideration of the inequalities of access to critical agricultural resources such as land, water, and inputs. Furthermore, the contrast between the CSA goals of increased productivity, resilience, and mitigation have been mashed with the triple-win discourse. The failure to acknowledge the contradictions between the social and environmental systems in achieving agricultural intensification has made it difficult to understand the challenges in reconciling the contrasts between the goals of productivity, resilience, and mitigation [123]. Adequate consideration has not been given to the possibility of uneven distribution of the benefits and costs of agricultural transformation on different social groups [54]. These masked constraints in the concept of CSA can become future tensions in the agricultural system.

Although the concept of CSA includes trade-offs between goals and will not affect all social groups the same way, the issue of equity is rarely addressed in CSA discourse [124]. Furthermore, the definition given to CSA by the FAO does not wholly address the critical concerns of resource access and control [27,125,126]. Karlsson et al. [124] further argued that addressing the issue of equity in CSA discourse is needful for political and instrumental applications. It is also vital to consider equity based on needs when addressing CSA concerning the small-scale farming system [54]. Including equity in

analyzing CSA gives a more holistic implication for policy and application. Table 4 summarizes the challenges involved in CSA adoption and up-scaling in sub-Saharan agriculture.

**Table 4.** Challenges for CSA Adoption and Up-Scaling in Sub-Saharan Agriculture.

| Challenge | Literature Sources |
|---|---|
| Reconciling the contrasts among CSA goals | Makate et al. [54]; Taylor [123]; Karlsson et al. [124] |
| Uncertainty regarding which CSA goals to prioritize | Karlsson et al. [124]; IPCC [18]; Partey et al. [33] |
| Addressing equity in mainstreaming CSA | FAO [27]; Makate et al. [54]; Karlsson et al. [124]; Newell and Taylor [125] |

## 4. Conclusions

Climate-smart agriculture avails unique opportunities for simultaneously tackling food security and facilitating adaptation and mitigation benefits. Sub-Saharan African nations will particularly benefit from CSA, given their vulnerability to the changing climatic condition, their heavy reliance on agriculture for livelihoods, and the critical position agricultural sector holds concerning food security in those nations. Embracing CSA as an efficient and prompt climate change response is highly essential for building capacity and attaining food security and sustainable agriculture in sub-Saharan Africa.

However, there is a need for differential approaches in advocating the acceptance and up-scaling of CSA. The small-scale agricultural sector in sub-Saharan Africa is characterized by a heterogeneous population. As a result, a single uniform approach will not be adequate in improving CSA practices among small-scale farmers in sub-Saharan Africa. The implication of this is that strategies to strengthen CSA adoption should factor in group specificity instead of mainstreaming strategies or actions at a general level. Therefore, stakeholders should consider implementing structures or modifying the structures already in existence to accommodate the heterogenous attribute of small-scale farmers and avoid the possible challenges that could arise by assuming a homogenous characteristic. Furthermore, CSA development in sub-Saharan Africa also depends on the readiness of farming households and institutions at both national and regional levels to comprehend the multi-dimensional climate change challenges and the consequent self-mobilization for developing and implementing policies to respond to the challenges at suitable scales.

**Author Contributions:** V.O.A. formulated the research investigation during his doctoral studies under the supervision of M.S. and A.O. V.O.A. undertook the review and draft manuscript compilation. Research scrutiny and validation were carried out by M.S. and A.O. The outcome of the manuscript is the effort of all the authors.

**Funding:** The National Research Foundation of South Africa and The World Academy of Science (NRF—TWAS) through the NRF—TWAS African Renaissance Doctoral Scholarship (Grant UID 105460) and the University of Zululand Research and Innovation Office, were the primary sources of funding for this research.

**Acknowledgments:** The authors acknowledge the National Research Fund (NRF), The World Academy of Science (TWAS) and the University of Zululand Research and Innovation Office for providing financial assistance for this research project.

**Conflicts of Interest:** The authors attest that there are no conflicts of interest.

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
