# Peer review of "The Dynamics of Climate Change Adaptation in Sub-Saharan Africa: A Review of Climate-Smart Agriculture among Small-Scale Farmers"

_climate, doi:10.3390/cli7110132_

Round 1

Reviewer 1 Report

This paper proposes a systematic review on climate-smart agriculture in Africa. The effort is appreciated, and such information for Africa is much welcome. I admit that the paper is not characterized by scientific novelty, and, as I explain in detail below, the systematic approach (as proposed) weakens the paper too much because of the concerns rising. In addition to what I mention below, Climate requires systematic reviews to follow the PRISMA guidelines, which are not followed in this paper. However, I also admit that, despite the science level of this paper is low, the information it provides is important and interesting for researchers working with environmental change as well as stakeholders, and with practical relevance. The paper is well written in general. Below I provide some comments that may help to improve the paper and overcome the limitations related to the systematic approach that I mention. All in all, this “review” paper is a concise summary of an interesting topic, and my recommendation is consideration for publication following revision.

The title is fine. The Abstract is also interesting and attractive.  So is the Introduction, which introduces in a concise and efficient way the background of the study as well as the aims of this paper. I have only a minor comment on this Section:

Lines 69-76: several interesting and promising biological means for enhancing plant productivity and enhancing agricultural sustainability under changing environments have been proposed in the literature ("see several papers on agricultural sustainability in the journal Global Food Security"). The potential applicability of such concepts within the framework of climate-smart agriculture in the African region should be considered in the Discussion of this paper.

Materials and Methods: Since the authors claim this is a systematic review, the methodology is of particular importance. However, I do not see anywhere any description about the search engines. It is necessary to state which search engines were used, and when the last search was done. It is also necessary to clarify what the search method was. Authors must also write all the keywords they have used. Furthermore, it is important to write the number of papers traced for each search, combination of keywords, and search engine. What were the exclusion criteria? They must be all clearly stated.

Results and Discussion: My major concern here is that I do not see any strong support for a systematic review. The paper would be more suitable for publication, and I would endorse it, if the authors do not “propose” the non-well supported systematic approach. This section makes me also curious why several other factors that may modify climate effects, such as air pollution, have been totally excluded. Another major concern is that there is no figure or table summarizing some interesting results/discussion. A minor comment here: The title of the section 3.1 does not match with the content of the section which eventually introduces only some climate patterns and nothing about agriculture.

Conclusions: This section promotes the concept of climate-smart agriculture in Africa, and it may be all right.

Author Response

.

Reviewer 2 Report

The article is well written but there are a number of areas to improve on:

First, the usage of the various concepts that underpin the results e.g., resilience, adaptation and adaptive capacity has to be consistent with global practice .

Second, the authors need to revisit the method section. It appears this is not thoroughly handled. My comments are returned in the attached manuscript.

Lastly some relevant literature on the subject matter is not reviewed. See my comments in the manuscript.

Author Response

.

Reviewer 3 Report

Kindly conduct another proof read of the text – normal things, but ensuring clarity and accuracy improves the text and is worth another cover-to-cover read through.

Methods: The methods are not actually clear – what search platforms were used? Web of Science? Google Scholar? Others? Multiple? I put “climate change adaptation” + “Africa” into Web of Science and got 4,000+ matches. It is unclear how the authors only arrived at 1,200 given that many more terms and combinations were used. At present, there is no way to replicate the study as we do not know the tools used. The selection of keywords also seems problematic; if CSA is focal to the study but emerged in 2010 as a concept, is the review from that point forward? The time parameters are not clear, what is the role of literature from 1945 to 2010?

Methods: Over 90% of results were excluded – this should be elaborated upon. Was something wrong with the search terms? Why so many false positives? This needs to be explained. This becomes clear when reading the short thematic sections – a wealth of research on climate change is covered in three references. If it is the objectives of CSA, as is implied, which exactly? And how was this applied to exclude so much of the literature?

I suggest the authors return to Berrang-Ford et al. or rethink how this paper is presented. At present, this might be framed as a synthesis drawing upon a subset of the literature. It currently does not appear to be representative of a systematic review of the literature.

Methods: No limitations? Suggest reflecting on the limitations and being transparent about those.

There is a suggestion that adaptation efforts have overly focused on technical solutions; however there is a large literature that does not. Consider one large set of research and pilots, for example over 40 academic papers (amongst many other outputs) here: http://www.assar.uct.ac.za/assar-outputs There are regional scale projects, as described here: https://link.springer.com/article/10.1007/s10113-017-1140-6 much of which took an interdisciplinary approach to adaptation. The review should really have captured all this literature. This should have been included under the “climate change adaptation” search terms – hence my questioning of the methods above.

The findings are frankly a bit hard to assess given the above methodological questions. For example, are the projects in 3.3 representative or purposively selected? The implementation of CSA also experiences a wide range of challenges, which are not covered. For example, CSA oriented projects advocate switching from chemical fertilizers to localized composting systems, however the trade-off is less fodder for livestock feed and in most cases insufficient supply. This requires bigger picture questions of land size, fragmentation, population, etc, not just a set of potentials – often made possible by external project support that is nearly impossible to take to national scale. Other projects (e.g. the Tanzania one) suggest an infrastructural component related to irrigation – since the paper did not define how broad or narrow CSA is being conceptualized, it is unclear what types of irrigation infrastructure are being included here – large infrastructural projects have significant GHG implications and can have negative impacts on watersheds and marine environments. If conceptually broad – what is the difference between CSA and traditional agricultural development? Irrigation has long-been promoted and critiqued, and has a long history of success and failure.

I was expecting to see a greater critical engagement with the literature, as opposed to high level descriptive summation. For example, why were these excluded?

Taylor, M. (2018). Climate-smart agriculture: what is it good for?. The Journal of Peasant Studies45(1), 89-107.

Karlsson, L., Naess, L. O., Nightingale, A., & Thompson, J. (2018). ‘Triple wins’ or ‘triple faults’? Analysing the equity implications of policy discourses on climate-smart agriculture (CSA). The journal of peasant studies45(1), 150-174.

Newell, P., & Taylor, O. (2018). Contested landscapes: the global political economy of climate-smart agriculture. The journal of peasant studies45(1), 108-129.

Pimbert, M. (2015). Agroecology as an alternative vision to conventional development and climate-smart agriculture. Development58(2-3), 286-298.

Arakelyan, I., Moran, D., & Wreford, A. (2017). Climate smart agriculture: a critical review. In Making Climate Compatible Development Happen (pp. 66-86). Routledge.

These seem highly relevant – but not featured. Leaves readers questioning the methods, or how a systematic review would have missed these, with the keywords in the title of the papers.

Author Response

.

Round 2

Reviewer 1 Report

The authors have significantly improved the paper taking into account all the comments and suggestions. The additions are very useful, and provide additional information that may be critical for the practices in the region. I have no objection to the additions, and the paper may be accepted for publication in its current form.